# Multiwalled Carbon Nanotubes Induce Fibrosis and Telomere Length Alterations

**DOI:** 10.3390/ijms23116005

**Published:** 2022-05-26

**Authors:** Mayes Alswady-Hoff, Johanna Samulin Erdem, Mona Aleksandersen, Kristine Haugen Anmarkrud, Øivind Skare, Fang-Chin Lin, Vincent Simensen, Yke Jildouw Arnoldussen, Vidar Skaug, Erik Ropstad, Shanbeh Zienolddiny-Narui

**Affiliations:** 1National Institute of Occupational Health, 0033 Oslo, Norway; mayes.alswady-hoff@stami.no (M.A.-H.); johanna.samulin-erdem@stami.no (J.S.E.); kristine.haugen.anmarkrud@stami.no (K.H.A.); oivind.skare@stami.no (Ø.S.); fang.chin.lin@stami.no (F.-C.L.); vincent.simensen@stami.no (V.S.); yke_arnoldussen@bio-rad.com (Y.J.A.); vskaug@online.no (V.S.); 2Faculty of Veterinary Medicine, Norwegian University of Life Sciences, 1432 As, Norway; mona.aleksandersen@nmbu.no (M.A.); erik.ropstad@nmbu.no (E.R.)

**Keywords:** manufactured nanomaterials, MWCNT, telomere instability, long-term exposure, inflammation, fibrosis, mesothelial hyperplasia, carcinogenicity

## Abstract

Telomere shortening can result in cellular senescence and in increased level of genome instability, which are key events in numerous of cancer types. Despite this, few studies have focused on the effect of nanomaterial exposure on telomere length as a possible mechanism involved in nanomaterial-induced carcinogenesis. In this study, effects of exposure to multiwalled carbon nanotubes (MWCNT) on telomere length were investigated in mice exposed by intrapleural injection, as well as in human lung epithelial and mesothelial cell lines. In addition, cell cycle, apoptosis, and regulation of genes involved in DNA damage repair were assessed. Exposure to MWCNT led to severe fibrosis, infiltration of inflammatory cells in pleura, and mesothelial cell hyperplasia. These histological alterations were accompanied by deregulation of genes involved in fibrosis and immune cell recruitment, as well as a significant shortening of telomeres in the pleura and the lung. Assessment of key carcinogenic mechanisms in vitro confirmed that long-term exposure to the long MWCNT led to a prominent telomere shortening in epithelial cells, which coincided with G1-phase arrest and enhanced apoptosis. Altogether, our data show that telomere shortening resulting in cell cycle arrest and apoptosis may be an important mechanism in long MWCNT-induced inflammation and fibrosis.

## 1. Introduction

Multiwalled carbon nanotubes (MWCNT) are classified as biopersistent high-aspect ratio nanomaterials (HARN) with at least one dimension below 100 nm. Due to their fibre-like shape and high durability, concerns have been raised regarding their potential health hazard as exposure to some HARN induce pulmonary and mesothelial pathologies, and tumours similar to the asbestos-like pathogenicity [1,2]. Malignant pleural mesothelioma, which is an aggressive cancer of the pleural membrane covering the lungs, is strongly linked to occupational exposure to asbestos [3]. Furthermore, workers exposed to asbestos and MWCNT show dysregulation in target genes linked to several pathways involved in cancer, fibrosis, and respiratory and cardiovascular diseases [3]. In 2014, the International Agency for Research on Cancer (IARC) classified the rigid and fibre-like Mitsui-7 (MWCNT-7) as possibly carcinogenic (group 2B; Monograph Volume 111) to humans based on evidence from animal and mechanistic experiments [4]. However, due to lacking data and large physiochemical heterogeneity, other types of carbon nanotubes were not classified. Data show that MWCNT may share several key characteristics of carcinogenesis such as oxidative stress, chronic inflammation, epigenetic alterations, genotoxicity, cellular immortalisation, altered cell proliferation, and cell death mechanisms, as reviewed in the work of Barbarino et al. [3]. Animals exposed to some MWCNT develop pulmonary inflammation, fibrotic lesions, and tumorigenesis [3,5,6,7]. Furthermore, MWCNT exposure in vitro may lead to increased levels of reactive oxygen species (ROS), trigger DNA damage, induce apoptosis, alter the cell cycle, and affect the secretion and production of inflammatory mediators such as IL8, TNF, IL6, and IL1B [8,9,10]. In addition to these well-established mechanisms, cellular senescence, which is known as an irreversible form of proliferative arrest, was recently identified as an emerging hallmark of cancer [11]. Fibrotic lung disease is believed to be mediated by senescent cells [12]. Moreover, cellular senescence can be caused by several cellular stress processes including telomere dysfunction [13].

Telomere structures at the end of linear chromosomes are important for preventing genome instability [13]. The maintenance of telomeres requires telomerase activity, which is a ribonucleoprotein complex consisting of telomerase reverse transcriptase (TERT), and a telomerase RNA component (TERC), in addition to a network of telomere-associated proteins [14]. Dysregulation in telomerase or other proteins and enzymes involved in the maintenance of the telomeres can result in abnormal telomere length and ultimately cell cycle arrest, senescence, telomere end fusion, and increased chromosome instability, which in turn is an important event in several cancer types [15,16]. Pulmonary fibrosis is associated with a reduction in telomere length [17]. Furthermore, it is believed that exposure to various occupational hazards may lead to telomere length alterations [18]. Limited studies have shown an association between MWCNT-exposure and telomere length; however, with inconclusive results. One study showed that exposure of human lung epithelial cells to MWCNT can result in telomere shortening and onset of cellular senescence [19]. Another study showed that long-term exposure of murine lung epithelial cells to MWCNT led to increased telomere length after 10 weeks of exposure [20]. Similarly, workers exposed to carbon nanotubes had increased telomere lengths in peripheral blood cells compared to the non-exposed control group [21]. Thus, more studies investigating telomere length alterations as a possible mechanism in MWCNT-induced fibrosis and carcinogenesis are warranted. This study aimed to investigate if the two MWCNT, Mitsui-7and NM-401, which share similar physical and chemical properties, may provoke similar effects involved in fibrosis and mesothelioma. Moreover, we aimed to investigate alterations in telomere length as a potential mechanism involved in MWCNT-induced inflammation, fibrosis, and carcinogenesis using both in vivo animal experiments and in vitro cell culture studies.

## 2. Results

### 2.1. Effects of MWCNT on Animal Growth and Histopathology

After week 14, MWCNT high dose exposed mice had decreased body weights compared to sham animals (Appendix A). The histological assessment showed accumulation of MWCNT fibres in exposed animals with severe histopathological pleural lesions, similar for both types of MWCNT. The lesions were multifocal and included prominent fibrosis, mononuclear cell infiltrates, granulomas, and hyperplasia of mesothelial cells (Figure 1). Exposed animals had significantly higher leukocyte infiltration and more fibrosis, granulomas, and mesothelial cell hyperplasia in the pleura when compared to the unexposed sham animals. Furthermore, exposed animals occasionally had focal and mild increase in interstitial fibrous tissue. When present, these changes were found in superficial parts of the lungs, underneath areas with severe pleural lesions. Histopathological lesions were not observed in the liver, spleen, or kidney of exposed animals (data not shown). The presence of granulomas was significantly severe in the high dose MWCNT groups compared to low dose and control groups (Table 1).

### 2.2. Effects of MWCNT on Genes Involved in Fibrosis and Inflammation

Intrapleural injection of Mitsui-7 and NM-401 manifested in alterations in the expression patterns of several genes involved in fibrotic processes. Overall, exposure to the two MWCNT resulted in very similar expression patterns in fibrosis genes, independent of dose (Figure 2a). Similar patterns were also observed for the lung and pleura, except for *Il10* and *Cdkn2a*, which were not expressed in the pleura. Of the 17 genes involved in fibrosis and inflammation investigated in this study, nine genes had the same pattern of regulation for both materials in pleura and lung of exposed animals, with the highest expression for *Ccl12* (high dose; pleura: 18-fold, lung: fourfold), *Cxcl2* (high dose; pleura: sevenfold, lung: twofold), and *Timp1* (high dose; pleura: Mitsui-7: eightfold, NM-401: 14-fold; lung: fourfold). Fold changes and *p*-values are shown in Appendix A. Furthermore, Mitsui-7- and NM-401-exposure resulted in an increased expression of *Ccl11* (high dose: twofold), *Cdkn2a* (high dose: fourfold), *Il10* (high dose: fourfold), *Col1a2* (high dose: sixfold), in the lung. Similarly, both materials resulted in significant upregulation of *Ccl3* (high dose: Mitsui-7: sixfold, NM-401: 10-fold), while the expression of *Timp4* was downregulated (high dose: 0.3-fold for both materials) in the pleura. Furthermore, *Bcl2* (0.3-fold) was only downregulated in Mitsui-7 low dose-exposed animals in pleura, while *Il1a* was significantly downregulated in Mitsui-7 low dose (0.4-fold)- and in NM-401 high dose (0.4-fold)-exposed animals in the lung. Selected genes involved in inflammation and fibrosis were run on HBEC-3KT and Met-5A exposed cells to NM-401 (Appendix A). Results from these selected genes supported the observation from animal experiment with a deregulation of genes involved in fibrosis and inflammation (Figure 2b). An upregulation was observed for *CXCL2* and *MMP2* in NM-401-exposed HBEC-3KT cells, as well as in the pleura and lung of exposed animals. Similarly, *CDKN2A* was upregulated following exposure of HBEC-3KT cells and in the lung tissue of exposed mice, while *CCL3* was upregulated following exposure of HBEC-3KT cells and in the pleura. Moreover, *IL8* expression was highly induced following NM-401-exposure in both HBEC-3KT (2000-fold) and Met-5A (fivefold) cells. However, none of the other genes analysed in this study were affected in NM-401-exposed Met-5A cells.

### 2.3. Effects of MWCNT-Exposure on Telomere Length

Telomere length in the pleura and lung of exposed animals and cells was determined by absolute qPCR (Figure 3a). In the pleura, a reduction in the telomere length of 2466 bp for low dose (*p* = 4.80 × 10^−2^) and 7374 bp for high dose (*p* = 8.00 × 10^−6^) was observed in Mitsui-7-exposed animals compared to sham. Moreover, telomere lengths in lung tissue were shortened by 8900 bp for low dose (*p* = 3.00 × 10^−3^) and a similar, however, not significant trend was observed for the high dose. Similarly, the telomere length was reduced by 4850 bp (low dose; *p* = 3.90 × 10^−8^) and 1729 bp (high dose; *p* = 1.10 × 10^−2^) in pleura of NM-401-exposed animals compared to sham animals. No effects of NM-401-exposure on the telomere length were observed in the lung tissues. On the other hand, in HBEC-3KT bronchial cells, NM-401-exposure led to a 1000 bp reduction in telomere length for both doses (Figure 3b). This reduction started already after 4 weeks and gradually increased until week 13 of exposure, with approximately 1200 bp shorter telomeres measured at week 13 for both doses. Accordingly, exposed cells had a higher percentage of short telomeres (<3000 bp), already at 4 weeks of exposure, and >60% of the telomers were shorter than 3000 bp at week 13 of NM-401-exposure (Figure 3c). In Met-5A cells, slightly longer telomeres (400 bp) were observed after 4 and 8 weeks of NM-401 high dose exposure, but no changes were observed after 13 weeks.

### 2.4. Effects of NM-401 on Pathway-Based Gene Expression Patterns in HBEC-3KT Cells

To study the effects of NM-401-exposure on HBEC-3KT cells, the expression of key regulatory genes important in the development of carcinogenesis were measured after 4, 8, and 13 weeks of exposure. Interestingly, a time-dependent regulation of genes involved in inflammation, cell cycle, and cell death mechanisms, such as *IL8*, *CXCL2*, *DRAM1*, *CYCS*, and *SOD2*, was observed following 13 weeks of NM-401-exposure (Figure 4a). Fold changes and *p*-values are shown in Appendix A. *IL8* was highly upregulated following NM-401-exposure, with a 2430-fold (low dose) and a 4027-fold (high dose) increase in expression at week 4, and a subsequent smaller increase in expression at week 13. Similarly, the expression level of *CXCL2* was 30-fold (low dose) and 45-fold (high dose) increased at week 4, and threefold (low dose) and sixfold (high dose) enhanced at week 13 of exposure. Moreover, *DRAM1* expression was 10-fold (low dose) and 12-fold (high dose) upregulated at week 4, and fivefold upregulated for both doses at week 13 of exposure. *SOD2* also showed a similar expression pattern, with an initial 12-fold upregulation at week 4 and 8, and a subsequent sevenfold upregulation at week 13 of exposure, independent of dose. Additionally, genes involved in cell death mechanisms were deregulated following MWCNT-exposure. Accordingly, *BAX* was found upregulated independent of dose at all times measured, while *CYCS* was downregulated independent of dose at week 8 of exposure (Figure 4b). In addition, we observed an upregulation in *TERT* at both doses and all time points and a downregulation in *TERC* from week 8 independent of dose (Figure 4b).

### 2.5. Effects of NM-401-Exposure on Cell Cycle and Apoptosis in Lung Epithelial Cells

Changes in cell cycle and apoptosis were assessed at week 4, 8, and 13 in NM-401-exposed cells. NM-401-exposure led to an increase in cells in the G1-phase at week 13, independent of dose (Figure 5a), while no changes were observed at 4 and 8 weeks of exposure. No effects were observed in Met-5A cells (Figure 5b). Furthermore, NM-401- exposure led to an increase in the number of apoptotic/necrotic HBEC-3KT cells, independent of dose and time, while a smaller increase in apoptotic/necrotic fraction of Met-5A cells was observed at week 4 and 13 only (Figure 5c).

## 3. Discussion

Several studies have reported evidence for development of fibrosis and cancer in animals exposed to some carbon nanotubes [9]. Accordingly, exposure to MWCNT leads to DNA damage, disruption of normal cell proliferation, cell cycle, and induction of genomic instability [9,10]. In this study, effects on fibrosis were assessed following exposure to the long and rigid MWCNT Mitsui-7 and NM-401 by intrapleural injection in mice. While this exposure model is not representative for pulmonary exposure, it is a known experimental method used to investigate pathological effects of fibres in the pleural space [22,23]. Here, both Mitsui-7- and NM-401-exposed animals showed accumulation of the MWCNT fibres in the pleura with accumulation of inflammatory cells and prominent fibrosis. In addition, we observed mesothelial hyperplasia in exposed animals, which is considered a precursor stage towards mesothelioma [24]. The malignant pleural mesothelioma, an aggressive cancer of the pleural membranes, has been strongly linked to asbestos exposure [3]. The pleural histopathological changes observed in the two MWCNT-exposed animals were associated with deregulation of genes involved in fibrosis and inflammation such as an upregulation of C-C motif chemokine 12 (*Ccl12*), eotaxin (*Ccl11*), and metalloproteinase inhibitor 1 (*Timp1*). Sustained inflammatory response following carbon nanotube exposure is thoroughly described in the work of Kuempel et al. [9]. Thus, while only Mitsui-7 has been classified as possibly carcinogenic [4], our data indicate that other long MWCNT of similar physiochemical properties and aspect ratio may hold similar potential carcinogenic properties. Moreover, NM-401-exposed HBEC-3KT cells revealed alteration in expression of several genes involved in fibrosis and inflammation, such as tumour necrosis factor (*TNF*), interleukin 8 (*IL8*), C-X-C motif chemokine 2 (*CXCL2*), gelatinase (*MMP2*), and C-C motif chemokine 3 (*CCL3*). In agreement with our results, *IL8*, which is an early pro-inflammatory biomarker, was highly expressed during nanomaterial-induced inflammation [10,25]. Previous studies reported that exposure to MWCNT induced persistent and chronic inflammation and fibrotic lesions in lungs, which coincide with deregulation of a number of genes associated with hallmarks of cancer [3,25,26]. However, while NM-401-exposure provided similar effects in animals and HBEC-3KT cells, no effects were observed in the mesothelial Met-5A MWCNT-exposed cells, except for an upregulation of *IL8*. Difference in deposited dose between the two cell lines could be a contributing factor as the elemental carbon measured in the low dose of HBEC-3KT exposed cells corresponded to the high dose of exposed Met-5A cells, and with higher number of seeded cells of Met-5A in mind, it results in much lower deposited dose compared to HBEC-3KT cells.

While inflammation, fibrosis, oxidative stress, DNA damage, cell proliferation, cell cycle, and genomic instability are acknowledged as well-established mechanisms of MWCNT-induced carcinogenesis, cellular senescence which generally is defined as a stable cell cycle arrest has been linked to MWCNT-exposure [27,28]. Increased cellular senescence may be linked to telomere shortening, which is an important event involved in carcinogenesis and in fibrotic pulmonary disease [11,13]. To clarify the involvement of telomere length alterations in MWCNT induced pathological and molecular alterations, the telomere length was measured in exposed animals to the MWCNT Mitsui-7 and NM-401 and in long-term NM-401-exposed cells in vitro. We observed that telomere length was reduced significantly in the pleural and lung tissues of both MWCNT-exposed animals. This effect was also observed in the long-term NM-401-exposed HBEC-3KT cells which had prominent telomere shortening and significantly higher percentage of telomeres shorter than 3 kbp compared to the controls. Loss of telomeres may lead to increased levels of recombination at chromosome ends, fusion of chromosomes, instability of the genome, growth arrest, and cell death [29]. The instability in telomere length is associated with increased cancer risk and other diseases such as pulmonary fibrosis [17,30]. Indeed, occupational exposures may disrupt telomere maintenance [18]. Exposure to welding fume and silica have been shown to affect the telomere length [18]. Moreover, asbestos exposure may cause telomere shortening in pleural mesothelial cells [31]. Similarly, TiO_2_ and vegetable carbon have been reported to induce telomere shortening in lung tissue and in epithelial cell lines [32,33]. Furthermore, shorter telomeres have been observed in A549 cells exposed to the MWCNT NM-400 after 48 h of exposure [19]. Our data are in concordance with these studies, as long-term exposure to MWCNT resulted in shorter telomeres in both animals and in pulmonary epithelial cells. On contrary, longer telomeres have been observed both in FE1-Muta^TM^Mouse lung epithelial cells [20] and in peripheral whole blood from MWCNT-exposed workers [21]. Together, our study and previous studies indicate that exposure to MWCNT leads to telomere alterations, but the exact effect and the mechanisms behind this are still not clear and require further investigation. To our knowledge, this is the first study showing that telomere shortening may be an important mechanism in MWCNT-induced fibrosis.

Critically short telomers result in activation of DNA damage response proteins, such as the phosphorylated forms of ataxia telangiectasia mutated (ATM), H2A histone family member X (H2A.X), and RAD17 checkpoint clamp loader component (RAD17) [34]. While many studies have shown that exposure to MWCNT leads to increased levels of DNA strand breaks and increased DNA repair activity and DNA damage [9,19,25], other studies did not observe genotoxicity following MWCNT-exposure [5,35,36]. In this study, we did not observe a regulation of genes involved in DNA damage or repair (i.e., *ATM*, *ATR*, *RAD17*, *NIEL3*, *XPA*, *ALKBH1*, *ALKBH5*, or *OGG1*) in HBEC-3KT cells exposed to NM-401. However, cells with critically short telomeres might reactivate the telomerase enzyme, allowing genome instability and immortalisation, which is a major hallmark of cancer [29]. In fact, the NM-401-exposed epithelial cells had an upregulation in *TERT*, but a downregulation of the *TERC* gene compared to unexposed cells. It is important to note that the HBEC-3KT cell line is a TERT immortalised cell line, and thus this mechanism requires further confirmation in other cell types. In mice, TERT or TERC deletion leads to telomere shortening, genomic instability, aneuploidy, telomeric fusion, and aging-related phenotypes [37,38,39]. However, overexpression of TERT induces constitutive telomerase activity and contributes to immortalisation in cells [40]. The downregulation of the *TERC* gene may explain the shortening in the telomeres observed in this study, but the upregulation in *TERT* gene remains unclear.

It has been widely acknowledged that telomere shortening results in senescence or apoptosis. However, when these mechanisms fail, carcinogenesis ensues [41]. There is also increasing evidence indicating that a single short telomere may be sufficient to start tumorigenesis, and telomerase activity might act upon only the shortest telomeres [30]. Recently, cellular senescence was included as a hallmark of cancer [11]. In this study, we observed a reduction in cell proliferation and altered cell cycle in NM-401-exposed HBEC-3KT cells. Accordingly, a G1-phase arrest was observed at week 13 of exposure to NM-401. This indicates that long-term exposure to NM-401 negatively affects the entry into S-phase, resulting in G1/S checkpoints activation. Furthermore, exposed HBEC-3KT cells had a gradual upregulation of cyclin-dependent kinase inhibitor 2A (*CDKN2A*) and a stable upregulation of cyclin dependent kinase inhibitor 1A (*CDKN1A*), suggesting cell cycle arrest in G1-phase. Similarly, an increased expression level of *Cdkn2a* was observed in exposed animals compared to sham animals. CDKN2A was significantly upregulated within lung samples in pulmonary fibrosis individuals, and this upregulation increased with disease severity [12]. An upregulation in CDKN2A is a classical feature of cellular senescence [42]. Cellular senescence is known for contributing to aging and tumour suppression, but in recent years, it has also been implicated in several pathological processes, including tumour promotion and lung fibrotic disease [12,43,44]. In addition, senescent cells may over time exhibit neoplastic conversion, senescence-associated secretory-phenotype-related chronic inflammation, and cell cycle re-entry of stemness-reprogrammed senescent cancer cells [45]. Recent studies have demonstrated that cellular senescence occurs as a response to MWCNT-exposure in human mesothelial cells and lung epithelial cells [27,28]. Furthermore, cells that reach critical short telomers may undergo apoptosis or cell senescence [46]. In fact, other studies revealed that exposure to MWCNT result in reduction of cell proliferation, cell arrest in G1-phase of cell cycle, and increased apoptosis [10]. In this study, we observed that the long-term exposure to NM-401 also increased the fraction of cells in late apoptosis/necrosis in HBEC-3KT. In agreement, an upregulation in *BAX* and *DRAM1* was observed in exposed cells compared to the control, while cytochrome c (*CYCS*) was downregulated. Although it is generally accepted that release of CYCS protein from the mitochondria is an important event in apoptosis, this could be prevented by a proteasomal degradation of CYCS in the cytosol [47].

In summary, our results indicate that animals exposed to MWCNT, Mitsui-7 and NM-401 developed fibrosis and infiltration of inflammatory cells in the pleura as well as mesothelial cell hyperplasia. This was accompanied by deregulation of genes involved in fibrosis and inflammatory responses. In addition, the pleural injection of MWCNT led to significant shortening of telomeres in the pleura and lung tissues in MWCNT-exposed animals. Further, in vitro long-term exposure to MWCNT NM-401 in lung epithelial cells led to shorter telomeres, which coincided with G1-phase arrest and enhanced apoptosis. These results may indicate that telomere shortening may be a potential mechanism involved in MWCNT-induced chronic inflammation and fibrosis.

## 4. Materials and Methods

### 4.1. Nanomaterials

Two high-aspect-ratio MWCNT were chosen for intrapleural injection, Mitsui-7 (MWCNT-7, NRCWE-006) and NM-401 (JRCNM04001a). The MWCNT were dispersed in DPBS (Mg^2+^ and Ca^2+^ free) supplemented with 5.5 mM D-glucose, 0.6 mg/mL serum albumin, 3% Tween-20, and 0.01 mg/mL 1,2 dipalmitoyl-sn-glycero-3 phosphocholine (DPPC) and sonicated on ice-bath at 30% duty cycles for 6 s on and 4 s off using a Branson SLPe probe sonicator (Branson Ultrasonic Corp., Danbury, CT, USA). The MWCNT used in this study were previously characterised by scanning electron microscopes (SEM) and by dynamic light scattering (DLS) [48,49]. Accordingly, Mitsui-7 fibres have a reported length of 5000 ± 4500 (nm ± SE) and diameter of 88 ± 5 (nm ± SE), NM-401 length of 4048 ± 2371 (nm ± SE), and diameter of 67 ± 24 (nm ± SE) [50,51]. Size distribution of Mitsui-7 measured by DLS had a hydrodynamic distribution of 1006 ± 42 nm (Polydispersity Index (PdI): 0.81 ± 0.02), while the hydrodynamic distribution of NM-401 was 1116 ± 54 nm (PdI: 0.79 ± 0.03) in an animal study. In vitro studies were performed with only NM-401 according to the NANOGENOTOX dispersion protocol [52]. NM-401 had a size distribution of 714 ± 82 nm (PdI: 0.35 ± 0.06) in dispersion media used for cell exposure. The hydrodynamic distribution of NM-401 nanomaterial differed in size between the different cell media, showing a higher length in the media used for HBEC-3KT cells (0 h: 1304 ± 307 nm, PdI: 0.40 ± 0.04; 72 h: 1466 ± 195 nm, PdI: 0.40 ± 0.07) than in media used for Met-5A (0 h: 682 ± 142 nm, PdI: 0.49 ± 0.1; 72 h: 772 ± 208 nm, PdI: 0.50 ± 0.05).

### 4.2. In Vivo Experiments

The study was approved by the Institutional Animal Care and Use Committee at the Norwegian University of Life Sciences (NMBU) and the Norwegian Food Safety Authority (application ID: 7503; Licence ID: 2015/60942-2). It was conducted in accordance with The Norwegian Regulation on Animal Experimentation at the Section for Experimental Biomedicine, NMBU-Faculty of Veterinary Medicine, in Oslo, Norway. The animals were health-monitored according to recommendations by the Federation of European Laboratory Animal Science Association (FELASA) and kept under specific-pathogen-free (SPF) conditions.

#### 4.2.1. Intrapleural Injection of the MWCNT

C57BL/6 mice were exposed by intrapleural injection in groups of 19 animals as described in our previous study [7]. Briefly, the animals were injected under anaesthesia with 100 µL of dispersed MWCNT to the achieved doses of 5 µg/mouse (low dose) or 50 µg/mouse (high dose). Sham exposed animals were injected to the equivalent volume of dispersion medium. The animals were kept for a total of 29 weeks, whereafter they were sacrificed using CO_2_. At sign of discomfort or illness, the mice were prematurely sacrificed.

#### 4.2.2. Collection and Preparation of Biological Samples for Histological and Molecular Analysis

Blood was collected from the vena cava inferior. Tissue specimens for histology were collected from pleura, lungs, liver, spleen, and kidneys; fixed in 4% formaldehyde; routinely processed; and embedded in paraffin. Tissue sections (3 μm) were stained with haematoxylin–eosin–safranin (HES) or haematoxylin and eosin (HE) and examined by light microscopy for (a) the presence of fibres, (b) leukocyte infiltration, (c) the proliferation of fibrous tissue, (d) mesothelial cell hyperplasia, and (e) the presence of granulomas. Lesions were assessed semi-quantitatively for each animal using a scoring system ranging from 0 (no changes) to 4 (severe changes). For gene expression analysis, lung and pleura samples were collected in RNAlater^®^ Stabilization Solution (Ambion, Thermo Fisher Scientific, Waltham, MA, USA) for RNA or DNA extraction.

### 4.3. In Vitro Cell Culture Experiments

Two human cell lines, a bronchial and a mesothelial cell line, were used. Human bronchial epithelial cells (HBEC-3KT, ATCC^®^ CRL-4051™) were obtained from ATCC, and the human mesothelial cells (MeT-5A, ATCC^®^ CRL-9444™) were received as a kind gift from Prof. Hannu Norppa (Finnish Institute of Occupational Health, Helsinki, Finland). Both cell lines were controlled for mycoplasma contamination and authenticated in 2020 and 2021 using DNA fingerprinting (Deutsche Sammlung von Mikroorganismen und Zellkulturen, Braunschweig, Germany). The HBEC-3KT cell line was maintained in a 1:1 mixture of LHC-9 (Gibco, Thermo Fisher Scientific; Waltham, MA, USA) and RPMI-1640 (Thermo Fisher Scientific). The Met-5A cell line was maintained in RPMI-1640 (Thermo Fisher Scientific) containing 10% FBS (Gibco). Both cell culture media contained 100 units/mL penicillin and 100 µg/mL streptomycin, and cells were maintained in a humidified 5% CO_2_ atmosphere at 37 °C. For cell experiments, 2.5 × 10^5^ (HBEC-3KT) and 5.0 × 10^5^ (Met-5A) cells were seeded in 152 cm^2^ cell culture plates (Sarstedt; Nümbrecht, Germany). The cells were exposed to 0.96 µg/cm^2^ (low dose) and 1.92 µg/cm^2^ (high dose) NM-401 twice a week for 13 weeks continuously, and sub-cultured once per week. Control (Ctrl) cells were exposed to the dispersion media only. Exposure experiments were performed once with two replicates in each exposure group. Cell proliferation and cytotoxicity were assessed using trypan blue and LDH-release assay (Lactate Dehydrogenase Activity Assay Kit; Thermo Fisher Scientific) (Appendix A). The doses were selected on the basis of the recommended exposure limit (REL) of 1 mg/m^3^ proposed by NIOSH (Cincinnati, OH, USA) [53]. The low dose was calculated to correspond to the lifetime dose accumulated by a worker 5 workdays/week, 48 weeks/year, for 20 years. The calculations were made on assumptions of an accumulation of breathing volume of 3 m^3^ in 8 h, a lung surface area of 140 m^2^, and around 30% alveolar deposition efficiency depending on particle size. The selected high dose is two times the low dose.

#### 4.3.1. Assessment of In Vitro Deposited Dose

NM-401 deposition was assessed by elemental carbon analysis essentially as described by Devoy et al. (2020) [54] with some modifications. In brief, 16,000 cells/well for HBEC-3KT and 32,000 cells/well for Met-5A were seeded in a 6-well plate and exposed twice for one week for NM-401 low or high dose. Cells were scraped in Solvable^TM^ (Perkin-Elmer; Courtaboeuf, France), incubated at 56 °C overnight, and centrifuged at 35,000× *g* for 10 min. The pellet was washed twice with MQ-H_2_O and transferred to a Pallflex 2500QAT-UP quartz filter (Pall Corporation, Port Washington, NY, USA). The filters were dried at 50 °C overnight, and elemental carbon content was assessed using a Sunset Laboratory Inc. OCEC Dual Optical Analyzer (Birch and Canary, NIOSH Method 5040). The instrumental method is based on the evaporation of organic carbon in a helium atmosphere followed by the subsequent oxidation of elemental carbon into CO_2_. The distinction between detected organic and elemental carbon was autoset and assured by incorporated calibration of the instrument with CH_4_ gas after every run. The elemental carbon levels were measured to 0.99 ± 0.26 µg/cm^2^ for low dose- and 1.63 ± 0.44 µg/cm^2^ for high dose- exposed HBEC-3KT cells. For Met-5A cells, elemental carbon was measured to 0.60 ± 0.12 and 1.05 ± 0.36 µg/cm^2^ for low- and high dose-exposed cells, respectively. Control cells had undetectable levels of elemental carbon.

#### 4.3.2. Assessment of Cell Cycle and Apoptosis

Flow cytometry was used to assess cell cycle and apoptosis after 4, 8, and 13 weeks of exposure to NM-401 nanomaterials. The cell cycle and the apoptosis including interference measurements were performed and analysed as described previously [32]. Briefly, cell cycle was determined by propidium iodide (PI) staining, while the apoptosis was detected by annexin V–APC conjugate and necrotic cells by PI staining (Invitrogen, Thermo Fisher Scientific). For both experiments, cells were filtered using 70 µm cell strainers (VWR, Radnor, PA, USA) to eliminate larger cell aggregates and particle aggregates/agglomerates. Furthermore, 1 × 10^6^ cells/mL were used to assess the different end points following standard protocols provided by the manufacturers unless otherwise stated. A sample run was performed using a CytoFLEX Flow Cytometer (Beckman Coulter, Brea, CA, USA), with 10,000 events recorded for further analysis. The FSC express 7 Flow Cytometry software (De Novo, Glendale, CA, USA) was used for gating and analysis. Small fragments and nanoparticles were gated out in the FSC/SSC scatter plot and excluded from further analysis. Nanoparticle interference with the flow cytometry analyses were assessed and data are included in Appendix A.

### 4.4. Gene Expression

RNA from homogenised lung and pleura tissue samples was extracted using an Animal Tissue RNA Purification kit (Norgen Biotek Corp., Thorold, ON, Canada), and from cells by a AllPrep DNA/RNA Mini Kit (Qiagen, Düsseldorf, Germany) according to the manufacturers’ instructions. For assessment of candidate gene expression profiles, RNA was reverse-transcribed using a qScript cDNA synthesis kit (Quanta BioSciences, Beverly, MA, USA) and expression analysed by real-time qPCR using Perfecta SYBR Green Fastmix Low ROX (Quanta Biosciences). qPCR analyses were performed on a QuantStudio 5 Real-Time PCR System (Applied Biosystems, Thermo Fisher Scientific). Gene expression analysis was normalised to the geometric mean of beta-2-microglobulin (B2M), glyceraldehyde 3-phosphate dehydrogenase (GAPDH), and 60S acidic ribosomal protein P0 (RPLP0). Relative expression was assessed using the ΔΔCT approach. CT values > 33 was set as non-detectable.

### 4.5. Telomere Length Assessment

DNA was extracted using an Animal Tissue DNA Purification kit (Norgen Biotek) from the homogenised lung and pleura tissue samples, and by an AllPrep DNA/RNA Mini Kit (Qiagen) from cells according to the manufacturers’ instructions. Absolute telomere length measurement was assessed using a standard curve approach by qPCR. The method was performed and analysed as described in the work of O’Callaghan et al. (2011) [55]. In addition, alterations of telomere length in HBEC-3KT cells exposed for 4, 8, and 13 weeks with NM-401 nanomaterials was verified by high-throughput Q-FISH at Life Length Company (Madrid, Spain), as previously described [32].

### 4.6. Statistical Analysis

In the animal study, gene expression fold change values and absolute values for telomere length measurements were log-transformed prior to analysis to ensure approximately normally distributed residuals. Each outcome was analysed by means of linear regression, adjusted for total exposure time and sex. The combined effect of dose and treatment on the outcome was studied using sham animals as reference. Changes in expression values > twofold and *p* < 0.05 were considered significant. The analysis was performed in R (version 4.1.2). The statistical analysis of the pathological data was performed using the Steel–Dwass non-parametric test. Long-term exposure study in vitro was only performed once, but with two biological replicates, due to practical limitations in performing such studies. Therefore, statistical analysis was not performed on data from the cell exposure studies. Figures were made using GraphPad Prism 9 software (GraphPad Software, San Diego, CA, USA).

## Figures and Tables

**Figure 1 ijms-23-06005-f001:**
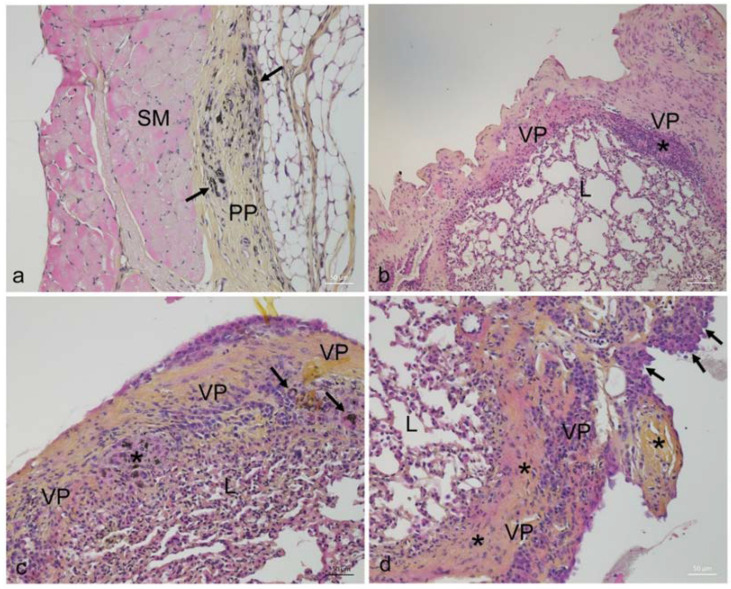
Pleural lesions in mice injected with MWCNT. (**a**) Thickening of parietal pleura with proliferation of fibrous tissue and multifocal infiltration of macrophages, multinucleated giant cells, plasma cells, and lymphocytes. Black arrows indicate macrophages and giant cells with visible MWCNT fibres. (**b**) Irregularly thickened visceral pleura with severe fibrosis. Asterisk indicates focal leucocyte aggregates. (**c**) Visceral pleura with proliferation of fibrous tissue and diffuse infiltration of inflammatory cells dominated by macrophages, multinucleated giant cells (arrows), and plasma cells. A granuloma with MWCNT-containing macrophages and giant cells is present (asterisk). An area with mild hyperplasia of mesothelial cells is found on the surface. (**d**) Severely thickened visceral pleura with proliferation of fibrous tissue (asterisks), mononuclear cell infiltrates, and hyperplasia of mesothelial cells (arrows). L: lung tissue, PP: parietal pleura, VP: visceral pleura, SM: skeletal muscle.

**Figure 2 ijms-23-06005-f002:**
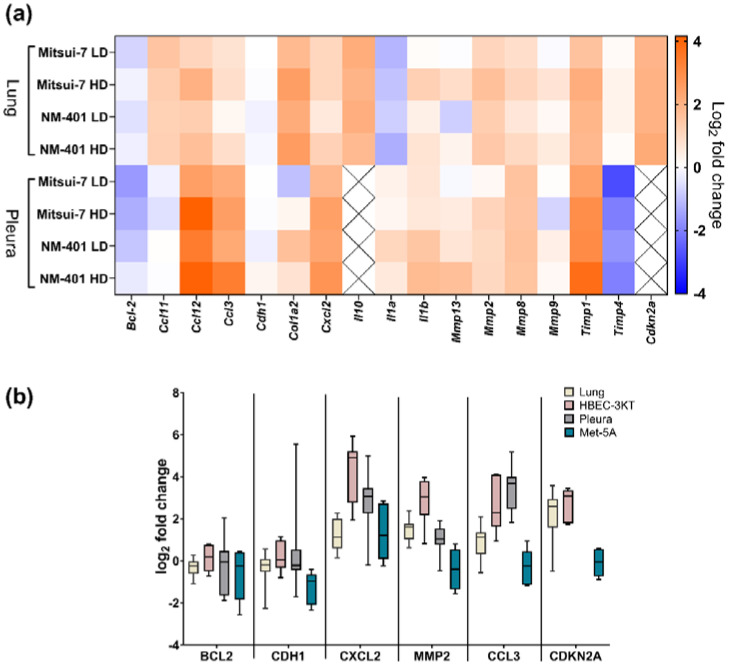
Effects of multiwalled carbon nanotubes (MWCNT)-exposure on genes involved in fibrosis and inflammation in vitro and in vivo. (**a**) Heatmap illustrating median fold changes in gene expression in pleura and lung following intrapleural injection of MWCNT Mitsui-7 and NM-401. Low dose (LD), high dose (HD); *n* = 12–18. (**b**) Selected genes following exposure to NM-401 high dose. Cellular data are merged data from all time points (*n* = 6). Data indicate mean ± 5 percentile.

**Figure 3 ijms-23-06005-f003:**
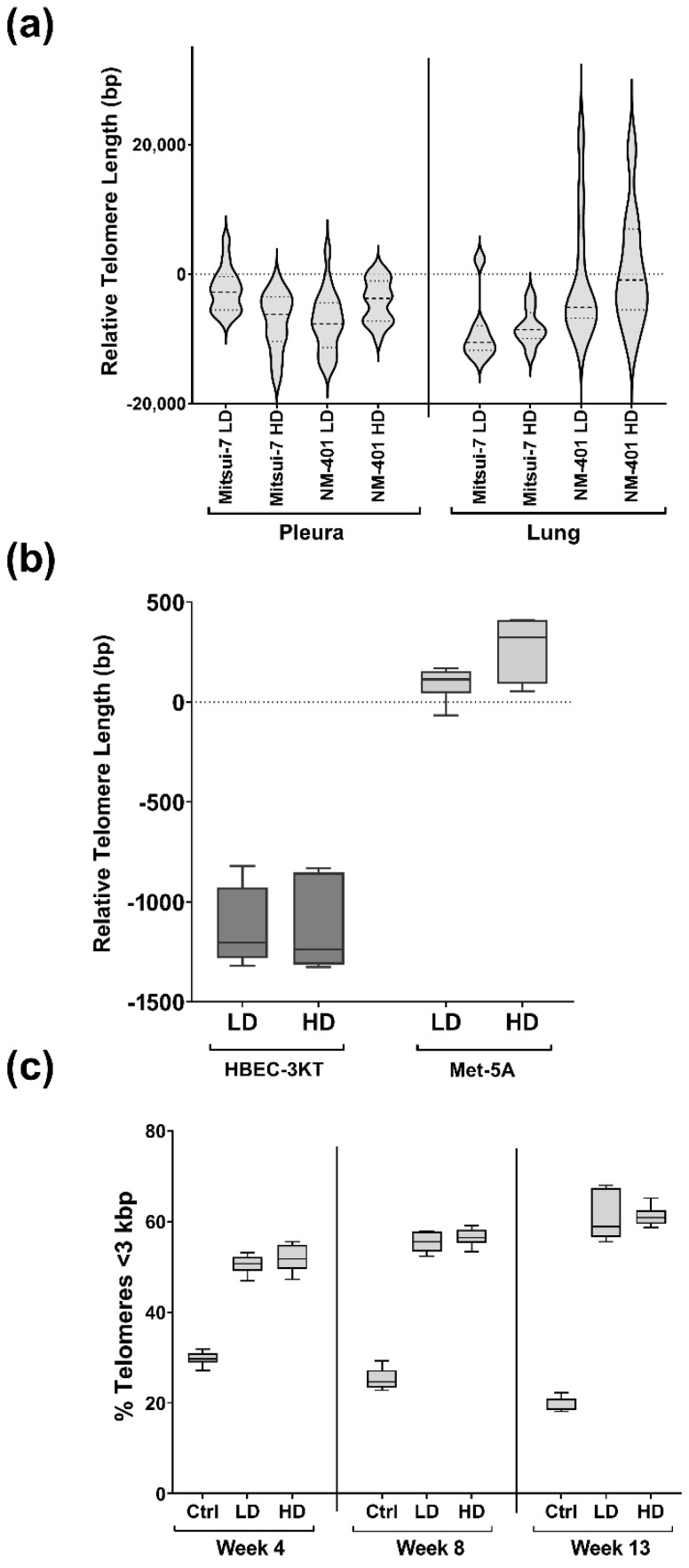
Effects of multiwalled carbon nanotubes (MWCNT)-exposure on telomere length. (**a**) Violin plots with relative telomere lengths in MWCNT Mitsui-7- and NM-401-exposed animals compared to sham animals (pleura: Mitsui-7 LD: *n* = 15 and HD: *n* = 14; NM-401 LD: *n* = 18 and HD *n* = 17; lung: Mitsui-7 LD: *n* = 7 and HD: *n* = 8; NM-401 LD: *n* = 10 and HD *n* = 9). (**b**) Box plot with relative telomere length in NM-401-exposed cells. Data indicates mean ± 5 percentiles of telomere length of merged data from all weeks measured (*n* = 6). (**c**) Percentage of telomeres in HBEC-3KT cells with a length below 3 kbp was measured after 4, 8, and 13 weeks of exposure to NM-401. Data indicate mean ± 5 percentiles (*n* = 2). Low dose (LD), high dose (HD).

**Figure 4 ijms-23-06005-f004:**
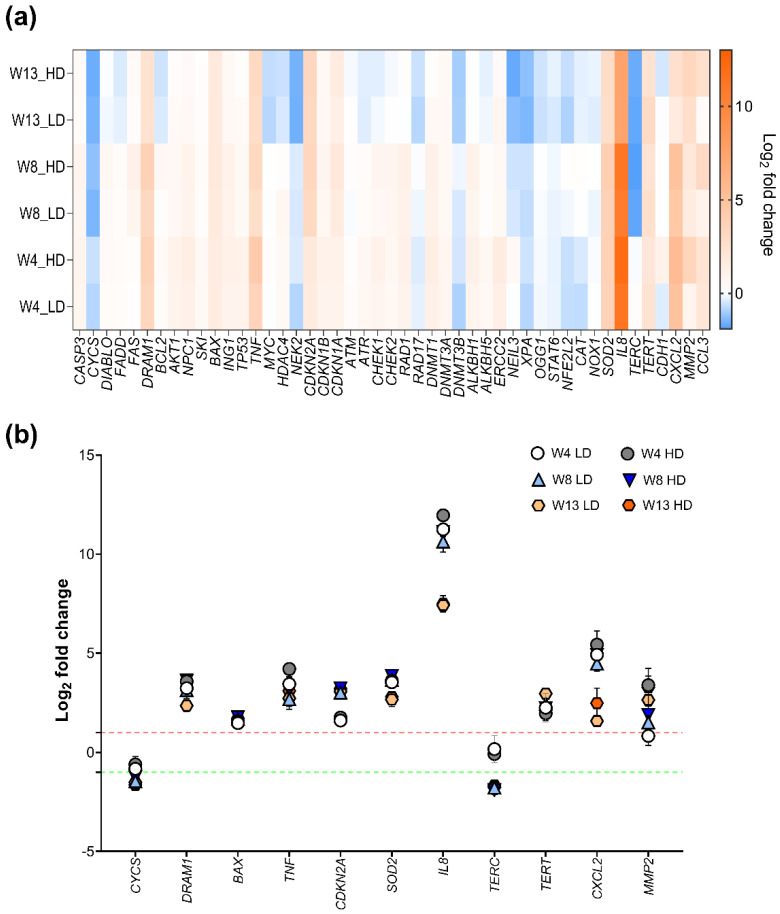
Regulation of gene expression in multiwalled carbon nanotubes (MWCNT)-exposed HBEC-3KT cells. (**a**) Heatmap illustrates mean fold changes in expression at 4 (W4), 8 (W8), and 13 (W13) weeks of NM-401-exposure. (**b**) Regulated genes at 4, 8, and 13 weeks of exposure to the LD and HD of NM-401. Low dose (LD), high dose (HD). Data indicates mean ± SD (*n* = 2). Dotted lines indicate log2 fold change 1 (red) and −1 (green).

**Figure 5 ijms-23-06005-f005:**
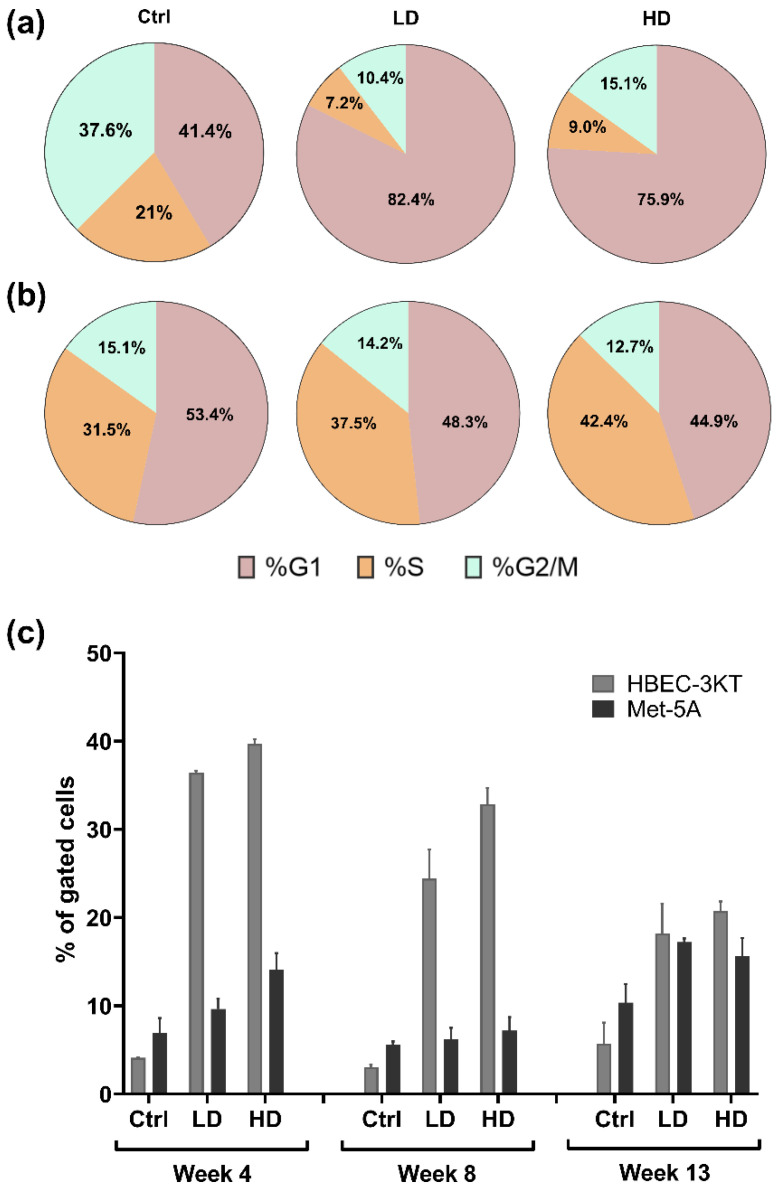
Effects of NM-401-exposure on cell cycle and apoptosis/necrosis. Cell cycle and apoptosis were assessed by flow cytometry. (**a**) Changes in cell cycle at week 13 of continuous exposure to NM-401 in HBEC-3KT cells. (**b**) Changes in cell cycle at week 13 of continuous exposure to NM-401 in Met-5A cells. (**c**) Apoptotic/necrotic cells at 4, 8, and 13 weeks of exposure to NM-401 in HBEC-3KT and Met-5A cells. Single cells were gated, and data are presented as % of gated cells. Low dose (LD), high dose (HD). Data indicate mean ± SD (*n* = 2).

**Table 1 ijms-23-06005-t001:** Effects of the MWCNT Mitsui-7 and NM-401 on morphological endpoints.

		Presence of Fibres	Infiltration of Leukocytes	Hyperplasia of Mesothelial Cells	Presence of Granulomas
Exposure	*n*	Mean ± SE	Mean ± SE	Mean ± SE	Mean ± SE
Sham	14	0 ^a^	0 ^a^	0 ^a^	0 ^a^
Mitsui-7 LD	15	1.57 ^b^ ± 0.15	2.47 ^b^ ± 0.11	2.03 ^b^ ± 0.12	0.00 ^a^ ± 0.00
Mitsui-7 HD	17	2.62 ^c^ ± 0.16	2.68 ^b^ ± 0.11	1.85 ^b^ ± 0.15	1.41 ^b^ ± 0.24
NM-401 LD	17	0.94 ^d^ ± 0.09	2.71 ^b^ ± 0.11	2.29 ^b^ ± 0.18	0.06 ^a^ ± 0.06
NM-401 HD	14	2.11 ^bc^ ± 0.13	2.82 ^b^ ± 0.10	1.89 ^b^ ± 0.22	1.15 ^b^ ± 0.27

Overall effects of fibre exposure on morphological endpoints for the presence of infiltration of inflammatory cells, granulomas, and fibrotic responses, in pleura of sham animals and mice exposed to multiwalled carbon nanotubes (MWCNT) Mitsui-7 or NM-401 at 5 or 50 μg/mouse (LD and HD, respectively). Results are means of semiquantitative scores (Grading 0–4). Levels not connected by the same letter (^a,b,c,d^) are significantly different (*p* < 0.05; the Steel–Dwass method). Low dose exposure group (LD); high dose exposure group (HD).

## Data Availability

Data are available from the researchers on request.

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
