# Peer review of "Multiwalled Carbon Nanotubes Induce Fibrosis and Telomere Length Alterations"

_ijms, 2022, doi:10.3390/ijms23116005_

Round 1

Reviewer 1 Report

In the manuscript, Alswady-Hoff M. and co-authors describe the effect of  Mitsui-7 and NM-401 MWCNTs in vivo (after intrapleural injection in  C57BL/6mice ) and in vitro (exposure to HBEC-3KT and MeT-5A cells). The authors demonstrated that exposed animals showed accumulation of the MWCNT fibers in the pleura with an accumulation of inflammatory cells and prominent fibrosis. Moreover, they observed a significant shortening of telomeres in treated animals' pleura and lung tissues.  Similar results were obtained in vitro experiments. The long-term exposure to NM-401 in epithelial cells led to shorter telomeres and coincided with G1-phase arrest and enhanced apoptosis. They suggest that MWCNT activated the above-described mechanisms inducing chronic inflammation and fibrosis.

The manuscript is well done and the results are well presented.

minor points

The authors have to clarify the different panels of genes among tissues and cells gene cells analyzed after treatment with Mitsui-7 and NM-401 MWCNTs.

The authors have to indicate in the text (lines 91-92) the nomenclature of samples, otherwise is difficult to understand the difference between HD and LD samples

Author Response

Reviewers Comments: ijms-1744280

“The authors have to clarify the different panels of genes among tissues and cells gene cells analyzed after treatment with Mitsui-7 and NM-401 MWCNTs”.

Reply: We thank the reviewer for pointing out an important detail that can be misunderstood in the results. To clarify this, we have modified the results section (page 4, line 131-134), and modified the figure 2 text (page 5).

Response to the reviewers’ comments

 “The authors have to indicate in the text (lines 91-92) the nomenclature of samples, otherwise is difficult to understand the difference between HD and LD samples”

Reply: We thank the reviewer for the comment. To avoid ambiguities, we replaced all HD and LD abbreviations in the text with “high dose” and “low dose” and the changes are marked in red.

Reviewer 2 Report

In this study, authors investigated variations of cell telomere lengths by an exposure of multi-walled carbon nanotubes (MWCNT). The exposure to MWCNT caused to severe fibrosis, and infiltration of inflammatory cell in pleura, mesothelial cell hyperplasia, G1-phase arrest and enhanced apoptosis. This article was well organized and sentences were cleared. If several contents were reinforced, they will be published in ` International Journal of Molecular Sciences’.

  1. When using an abbreviation for the first time, include your full name. (eg. MWCNT HD: multi-walled carbon nanotubes high dose (In general, HD MWCNT means highly dispersive multi-walled carbon nanotubes.) Similarly, In Table 1, a detailed description of each exposed MWCNT is required.

  1. To better explain the meaning of Figure 2, it is necessary to add a heat map by the gene expression using the two cells (HBEC-3KT, Met-5A) used in figure 2B.

  1. In figure 5 authors showed the effects of NM-401 exposure on cell cycle and apoptosis/necrosis in HBEC-3KT cells. Although, no effects were observed in Met-5A cells, figures should be supplied to the reader to help an understanding the manuscript.

  1. Authors only analyzed the high dose effect in Figure 4b) for 3 different weeks. However, the analysis of the effect of low dose, already presented in Figure 4a), should also be presented inside of the figure.

  1. Describe meanings of red dotted line and green dotted line in Figure 4b).

Author Response

Reviewer: 2

We thank the reviewer for the valuable comments.

1. When using an abbreviation for the first time, include your full name. (eg. MWCNT HD: multi-walled carbon nanotubes high dose (In general, HD MWCNT means highly dispersive multi-walled carbon nanotubes.) Similarly, In Table 1, a detailed description of each exposed MWCNT is required.

Reply: We thank the reviewer for the observant and specific valuable comments. We have now included the descriptions of abbreviations at first mention and have replaced all HD and LD abbreviations with “high dose” and “low dose” in the text to avoid misunderstandings. All changes made are marked in red.

2. To better explain the meaning of Figure 2, it is necessary to add a heat map by the gene expression using the two cells (HBEC-3KT, Met-5A) used in figure 2B.

Reply: As asked by the reviewer, we have now included a new heatmap in the supplementary section (Figure S2) to show the selected genes run for the two cell lines. Furthermore, to clarify this better we have modified the text in the results section on page 4, line 131-134, and modified the figure 2 text on page 5.

3.  In figure 5 authors showed the effects of NM-401 exposure on cell cycle and apoptosis/necrosis in HBEC-3KT cells. Although, no effects were observed in Met-5A cells, figures should be supplied to the reader to help an understanding the manuscript.

Reply: We have included Met-5A data in figure 5 (Fig. 5B and C) and modified the text accordingly, page 8, line 206 and 208-209.

4. Authors only analyzed the high dose effect in Figure 4b) for 3 different weeks. However, the analysis of the effect of low dose, already presented in Figure 4a), should also be presented inside of the figure.

Reply: We have included the low dose in the figure 4B as suggested by the reviewer and appreciate that inclusion of the low dose in figure 4b will make the results more easily understandable.

5. Describe meanings of red dotted line and green dotted line in Figure 4b).

Reply: We thank the reviewer for his observant comments. A sentence has been added to the figure 4 text (figure 4, page 8, line 200-201, marked in red).